# Spatial disinfection potential of slightly acidic electrolyzed water

**Midori Kurahashi[1]☯, Takaaki Ito[2]☯, Angelica Naka📷[1]☯ \***

**1** Graduate School of Agricultural and Life Sciences, The University of Tokyo, Tokyo, Japan, **2** Organo Food Tech Corporation, Saitama, Japan

☯ These authors contributed equally to this work.

\* angelica.naka@mail.ecc.u-tokyo.ac.jp

**Data Availability Statement:** All relevant data are within the paper and supporting information files.

**Funding:** This research was possible thanks to the annual research budget of the University of Tokyo. We would like to thank Organo Food Tech Corporation for providing the slightly acidic

## Abstract

Slightly acidic electrolyzed water (SAEW) was developed by Japanese companies over 20 years ago. SAEW has the advantage of potent sterilizing action while being relatively safe. This study evaluated the potential application of SAEW in spatial disinfection. Prior to experiments involving spatial spraying, the ability of SAEW to remove seven type of microorganisms that cause food poisoning was studied in vitro. Results indicated that free chlorine in SAEW, even at a low concentration (30 mg/L), was able to remove *Cladosporium cladosporioides*, a typical airborne fungus that degrades food, and spores such as *Bacillus subtilis*, a hardy bacterium. In an experiment involving spatial spraying, 3.43 $\log_{10}$ CFU/100 L of *Staphylococcus epidermidis* was sprayed in a room-sized space; the same space was then sprayed with SAEW. The number of settling microbes was measured and the sterilizing ability of SAEW was assessed. Results indicated that the concentration of *S. epidermidis* in the space was completely removed after 20 minutes of SAEW spraying. The above findings indicate that SAEW may be used to remove airborne microorganisms via spatial spraying.

## 1 Introduction

Hypochlorite-based and chlorine dioxide-based agents are widely used around the world as chlorine-based sanitizers. Commercially available sodium hypochlorite solutions have free available chlorine concentrations of 50,000–120,000 mg/L; this is typically diluted to 100–200 mg/L when used. Sodium hypochlorite solutions are inexpensive and relatively effective, so they have been used around the world for over 100 years. However, bromic acid can contaminate sodium hypochlorite solutions, as a by-product of the production process. Bromic acid is a chemical substance that recent research suggests is carcinogenic [1]. Moreover, certain interactions between sodium hypochlorite and organic matter can generate chloroform (a trihalomethane) [2], another chemical substance that is suspected of being carcinogenic [3]. A trade-off for the potent sterilizing action of chlorine dioxide is that the substance is extremely dangerous in its pure form, requiring a material safety data sheet (MSDS). Thus, chlorine dioxide is typically available as stabilized chlorine dioxide (actually "stabilized chlorite"). Stabilized chlorine dioxide is the general name given to a substance or mixture that generates chlorine dioxide rather than the actual name of a substance. Stabilized chlorine dioxide refers to any

electrolyzed water (SAEW) and the clean chamber to conduct spatial disinfection test. Organo Food Tech Corporation contributed to the design of the experiments and we discussed the results before writing the manuscript, but they had no role in data collection and analysis or decision to publish.

**Competing interests:** The author TI's commercial affiliation (Organo Food Tech Corporation) provided the SAEW and the clean chamber (resources) to conduct spatial disinfection experiments. There are no patents, products in development or marketed products to declare. This does not alter our adherence to PLOS ONE policies on sharing data and materials.

substance that is physically stable and adjusted to gradually release chlorine dioxide in accordance with environmental changes such as temperature and pH (the extent of release differs depending on the product). In addition, the potency of stabilized chlorine dioxide and pure chlorine dioxide differs vastly even at the same concentration. A vast variety of products is sold as stabilized chlorine dioxide, but the methods of stabilization vary depending on the product, so assessing their overall safety and effectiveness is difficult.

In Japan, companies have developed products by electrolyzing a dilute hydrochloric acid solution in a diaphragm-less electrolytic cell and diluting with tap water to a pH of 5–6.5. This aqueous solution is used as a sterilizing solution. A major sterilizing component in slightly acidic electrolyzed water (SAEW) is hypochlorous acid (HOCl), this is different from the hypochlorous acid solution produced by mixing an acid with sodium hypochlorite. Japan's Ministry of Health, Labor, and Welfare designated SAEW as a food additive sanitizer in 2002, and the Ministry of Agriculture, Forestry, and Fisheries and the Ministry of the Environment designated SAEW as a control agent in 2014. Thus, areas in which SAEW is used in Japan are expanding, rapidly gaining acceptance in the area of food sanitation in particular.

One of SAEW's potential applications may be spatial disinfection due to its safety and sterilizing capacity. Spatial disinfection with a conventional agent involves spraying in an unoccupied space or using protective equipment. The disinfected space generally requires proper ventilation afterwards. In contrast, SAEW has sterilizing action even at a low concentration. Its efficacy as disinfectant was measured in this research by directly applying SAEW on different types of bacteria and bacterial spores. Spatial disinfection can be performed spraying SAEW with free chlorine concentrations of 100 mg/L or lower and without concern for residual chlorine. There are numerous studies on the sterilizing action of electrolyzed water, but there are very few studies on the use of SAEW for indoor disinfection [4–6]. Given the emergence of multidrug-resistant microorganisms and the increased risk of infectious diseases due to climate change [7], the demand for disinfection of spaces such as food factories, hospitals, care facilities, kindergartens, and animal pens and coops is expected to continue growing in the future. The current study assessed disinfection when spraying SAEW in a space, and it discusses the potential for spatial disinfection with SAEW.

## 2 Materials and methods

### 2.1 Microbial culture and preparation of inocula

An in vitro experiment was performed with *Bacillus cereus* NBRC 13494, *Bacillus subtilis* NBRC 3134, *Pseudomonas aeruginosa* NBRC 13275, *Salmonella enterica* subsp. *enterica* NBRC 3313, *Staphylococcus aureus* subsp. *aureus* NBRC 12732, *Cladosporium cladosporioides* NBRC 6348, and *Staphylococcus epidermidis* NBRC 12993 from the NITE Biological Resource Center (NBRC, Japan) and *Escherichia coli* ATCC 43895 (serotype O157:H7, a verotoxin I and II-producing strain) from the American Type Culture Collection (ATCC, USA).

*B. cereus* was cultured on nutrient agar PT2810 (Eiken Chemical Co.) and *B. subtilis* was cultured on soybean-casein digest agar PT8010 (Eiken Chemical Co.) at 35°C±1°C for 7–10 d. Each bacterium was then suspended in physiologic saline solution, heated to 70°C±1°C for 20 min to kill nutritive cells. Each suspension was centrifuged, the supernatant was removed, bacterial cells were suspended in physiologic saline solution, and the spore count was adjusted to about $10^8$–$10^9$ CFU/mL to yield the bacterial spore solution. The bacterial spore solution was diluted with physiologic saline solution and adjusted to $10^7$–$10^8$ CFU/mL to yield the test bacterial solution. *E. coli*, *P. aeruginosa*, *S. enterica*, *S. aureus* were each cultured on nutrient agar PT 1010 (Eiken Chemical Co.) at 35°C±1°C for 18–24 h. Each bacterium was then suspended in physiologic saline solution and the bacterial count was adjusted to $10^7$–$10^8$ CFU/mL to yield

the test bacterial solution. *S. epidermidis* was cultured in a brain heart infusion broth PT3025 (Eiken Chemical Co.) at 35˚C±1˚C for 18–24 h. The bacterial count was adjusted to $10^7$–$10^8$ CFU/mL with a sterile phosphate buffer (phosphate-buffered saline, or PBS) to yield the test bacterial solution. *C. cladosporioides* was cultured on potato dextrose agar PT 2610 (Eiken Chemical Co.) at 25˚C±1˚C for 7–10 days. Spores were suspended in a 0.005% dioctyl sodium sulfosuccinate solution. The suspension was filtered through a non-woven fabric filter. The fungal count was adjusted to $10^7$–$10^8$ CFU/mL to yield the test fungal solution.

A spatial disinfection experiment was performed with *S. epidermidis* NBRC 12993 from the NITE Biological Resource Center (NBRC), Japan. The test bacterium was cultured in a brain heart infusion broth PT3025 (Eiken Chemical Co.) at 35˚C±1˚C for 18–24 h. The bacterial count was adjusted to $10^6$ CFU/mL with a sterile phosphate buffer (phosphate-buffered saline, or PBS) to yield the test bacterial solution. One mL of PBS was added to 2 mL of the bacterial solution to create 3 mL of spray solution. *S. epidermidis* was the only microorganism selected for spatial disinfection experiments because it is a common bacterium present in the human skin and, thus, does not usually cause infections in non-compromised patients and does not require high level of biosafety laboratories.

## 2.2 Preparation of treatment solutions

Slightly acidic electrolyzed water (HOCL0.36 t, HOCL Inc.) was specifically prepared by Organo Food Tech, Saitama, Japan for the purpose of these experiments. The SAEW used in experiments involving pure cultures had an available chlorine concentration of 30 mg/L and a pH of 5.9. SAEW used in the spatial disinfection experiment had an available chlorine concentration of 41 mg/L and a pH of 6.0.

## 2.3 Effects of SAEW on pure cultures

Experiment on pure cultures was prepared by mixing 9.9 mL of SAEW with 0.1 mL of the test microbial solution. This test solution was stored at 20˚C±1˚C. After 15 s, 1 min, and 15 min, it was immediately diluted 10-fold with soybean-casein digest medium with polysorbate 80 & lecithin SCDLP "DAIGO" (Nihon Pharmaceutical Co.). Microorganisms in the test solution were counted using a medium for cultivation and isolation. The medium for cultivation and isolation was soybean-casein digest agar with polysorbate 80 & lecithin SCDLP "DAIGO" (Nihon Pharmaceutical Co.) for *B. cereus*, *B. subtilis*, *E. coli*, *P. aeruginosa*, *S. enterica*, *S. aureus*, and *S. epidermidis*. Each bacterium was then cultured at 35˚C±1˚C for 2 days using the pour plate method. The medium for cultivation and isolation was glucose peptone agar with lecithin & polysorbate 80 GPLP AGAR "DAIGO" (Nihon Pharmaceutical Co.) for *C. cladosporioides*. The fungus was cultured at 25˚C±1˚C for 7 days using the pour plate method. Similar testing was performed using purified water as a control, and viable microorganisms were counted at the start and after 15 min. All experiments were performed in triplicate.

## 2.4 Effects of indoor spraying of SAEW

Negative pressure was generated in a clean chamber with a volume of 26 $m^3$ (length: 2.99 m, width: 3.96 m, height: 2.21 m), internal air was sampled with an air sampler (SAS Super ISO100, PBI, Milan, Italy), the purity of the air inside was measured, and air was determined to be sufficiently pure. *S. epidermidis* adjusted to $1×10^6$ CFU/mL was mixed with PBS. Three mL of this mixture (2 mL of the bacterial solution+1 mL of PBS) was sprayed at a rate of 1 mL/min for 3 min from the spray nozzle of an ultrasonic humidifier AHD-012 (Shizuku Plus+, Apix International, Osaka, Japan) in a large clean chamber. A fan for bacterial dispersal was run for 1 min; once the air in the clean chamber was agitated, it was sampled. This served

as 0 min. The ultrasonic humidifier was then run at 5.5 mL/min, and spraying of SAEW began. Air (100 L) was sampled with the air sampler every 10, 20, 30, and 40 min, and airborne bacteria were counted. Testing was similarly performed without running the ultrasonic humidifier and when spraying sterile water to serve as negative controls. This experiment was repeated three times.

## 2.5 Statistical analysis

All experiments were performed in triplicate. Microbial counts are the mean±standard deviation for individual samples. The obtained results were statistically assessed using Tukey's honestly significant difference (HSD) test, with $P < 0.05$ considered to be statistically significant.

# 3 Results

## 3.1 Effects of SAEW on pure cultures

Results of in vitro experiment involving typical microorganisms that cause food poisoning are shown in Fig 1. Treatment of *B. cereus* (spores), a gram-positive bacillus that causes food to spoil, resulted in 1/10 of the original $\log_{10}$ CFU/mL after 60 sec. After 15 minutes, a logarithmic decrease of 4.9 was observed. Similarly, after *B. subtilis* (spores) was treated with SAEW, a logarithmic decrease of 5.0 or greater ($\log_{10}$ CFU/mL) was noted within 15 min. After *E. coli* (O157:H7), *P. aeruginosa*, and *S. enterica*, which are typical gram-negative bacteria that cause food poisoning, were treated with SAEW, a respective logarithmic ($\log_{10}$ CFU/mL) decrease of 5.5, 5.8 and 6.2 or greater was noted within 15 min. Treatment of *S. aureus* and *S. epidermidis*, gram-positive facultative anaerobic bacilli, with electrolyzed water resulted in logarithmic ($\log_{10}$ CFU/mL) decrease of 5.2 or greater and 5.4 or greater, respectively within 15 sec. After *C. cladosporioides*, a typical contaminating fungus, was treated with SAEW, a logarithmic decrease of 5.7 or greater ($\log_{10}$ CFU/mL) was noted within 60 sec.

## 3.2 Effects of indoor spraying of SAEW

To determine the microbe count, 100 L of air was sampled 5 times every 10 min, so a total of 500 L was sampled. That air was supplied from outside and passed through a HEPA filter. Thus, a correction for that amount would need to be made to achieve accuracy. However, since only 1.92% of the chamber volume was sampled, no correction was performed in this study.

Fig 2 shows changes in the amount of airborne bacteria in the large clean chamber over time. Natural bacterial attenuation and humidity due to water spraying may impact bacterial concentration. Thus, gradual bacterial attenuation due to both, natural attenuation and water were also evaluated. After 20 minutes, the amount of the logarithmic decrease with natural attenuation was 0.54 $\log_{10}$ CFU/100 L, whereas the logarithmic decrease with sprayed water was 0.86 $\log_{10}$ CFU/100 L. Immediately after SAEW was sprayed, however, the bacterial count abruptly decreased, and it was below detection sensitivity after 20 min. The logarithmic decrease was 3.43 $\log_{10}$ CFU/100 L.

Test bacterium is dispersed by sprayed water, so the humidity in the chamber may impact the amount of the bacterium. In order to reduce the amount of moisture for a sprayed bacterium, an ultrasonic mister was used instead of a typical nebulizer. Changes in humidity in the large clean chamber under different conditions are shown in Fig 3. The chamber temperature during measurement was 24°C. Humidity under natural conditions was almost constant at 40%, but it increased from 40 to nearly 80% in 40 min when sterile water or SAEW was sprayed. Moreover, the effects of humidity on the amount of airborne bacteria can be

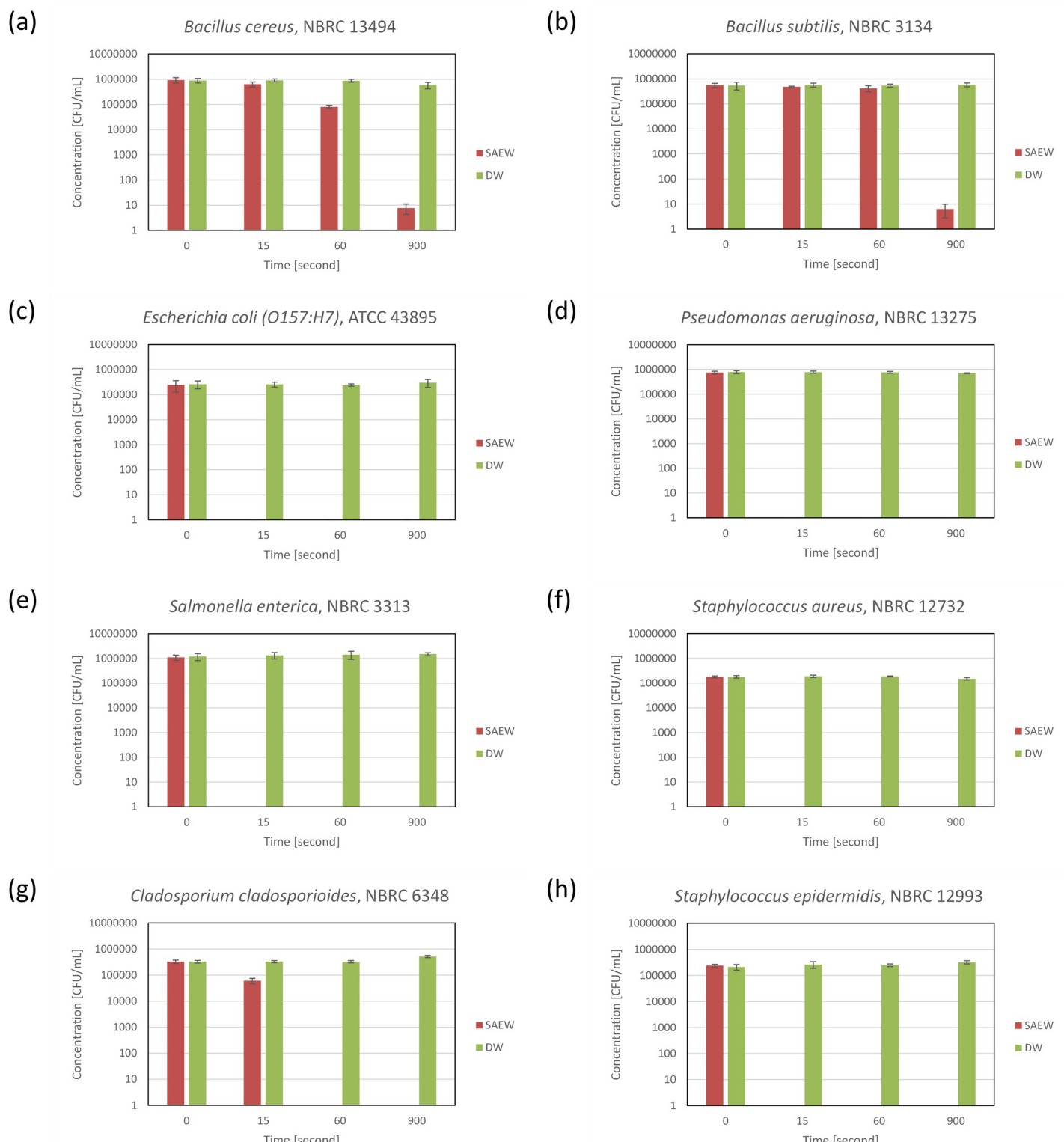

**Fig 1.** Effects of SAEW on pure cultures (a) *Bacillus cereus*, (b) *Bacillus subtilis*, (c) *Escherichia coli* (O157:H7), (d) *Pseudomonas aeruginosa*, (e) *Salmonella enterica subsp. enterica*, (f) *Staphylococcus aureus subsp. aureus*, (g) *Cladosporium cladosporioides*, and (h) *Staphylococcus epidermidis*.

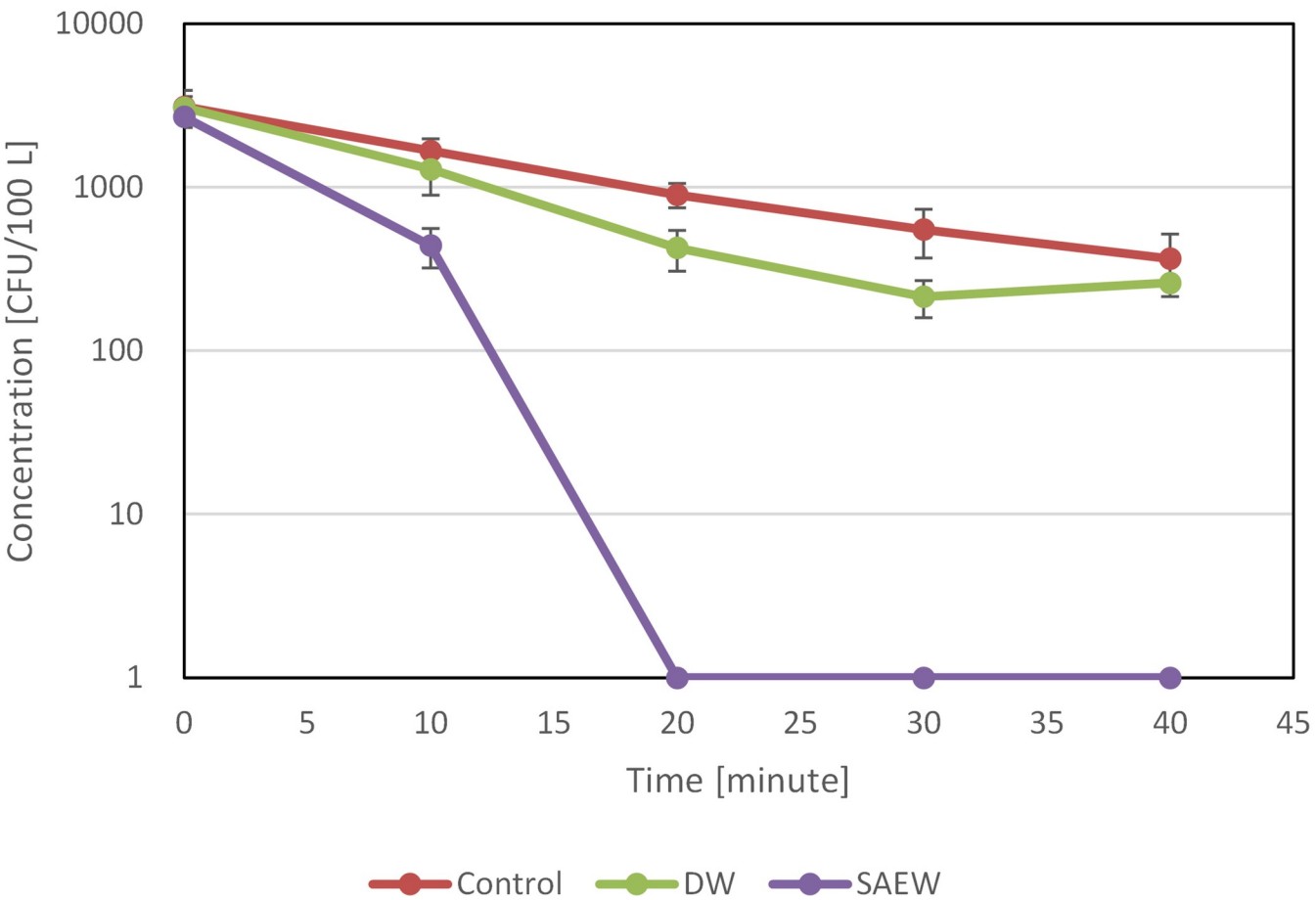

**Fig 2. Bactericidal effect of slightly acidic electrolyzed water spray on airborne bacteria.**

ascertained by comparing Figs 2 and 3. Fig 2 shows how the bacterial count gradually decreased with natural attenuation and the spraying of sterile water but it markedly decreased abruptly with the spraying of SAEW. In other words, this experiment revealed that the effects of humidity on the amount of airborne bacteria are extremely low compared to the effects of SAEW on the reduction in the bacterial count.

## 4 Discussion

Besides sodium hypochlorite and chlorine dioxide, aqueous solutions obtained by electrolyzing a sodium chloride solution are also used as chlorine-based disinfectants. When a dilute sodium chloride solution is electrolyzed in an electrolytic cell with a diaphragm, a strongly acidic aqueous solution is generated in the anode chamber and a strongly alkaline aqueous solution is generated in the cathode chamber. Strongly acidic electrolyzed water (AEW) generated in the anode chamber has potent sterilizing action [8–13]. However, the sterilizing action of AEW disappears in a short period of time [14]. In addition, alkaline water is generated in the cathode chamber in the same volume as strongly acidic electrolyzed water in the anode chamber. Thus, only half the volume of the sodium chloride solution (the stock solution) is obtained as a sterilizing solution.

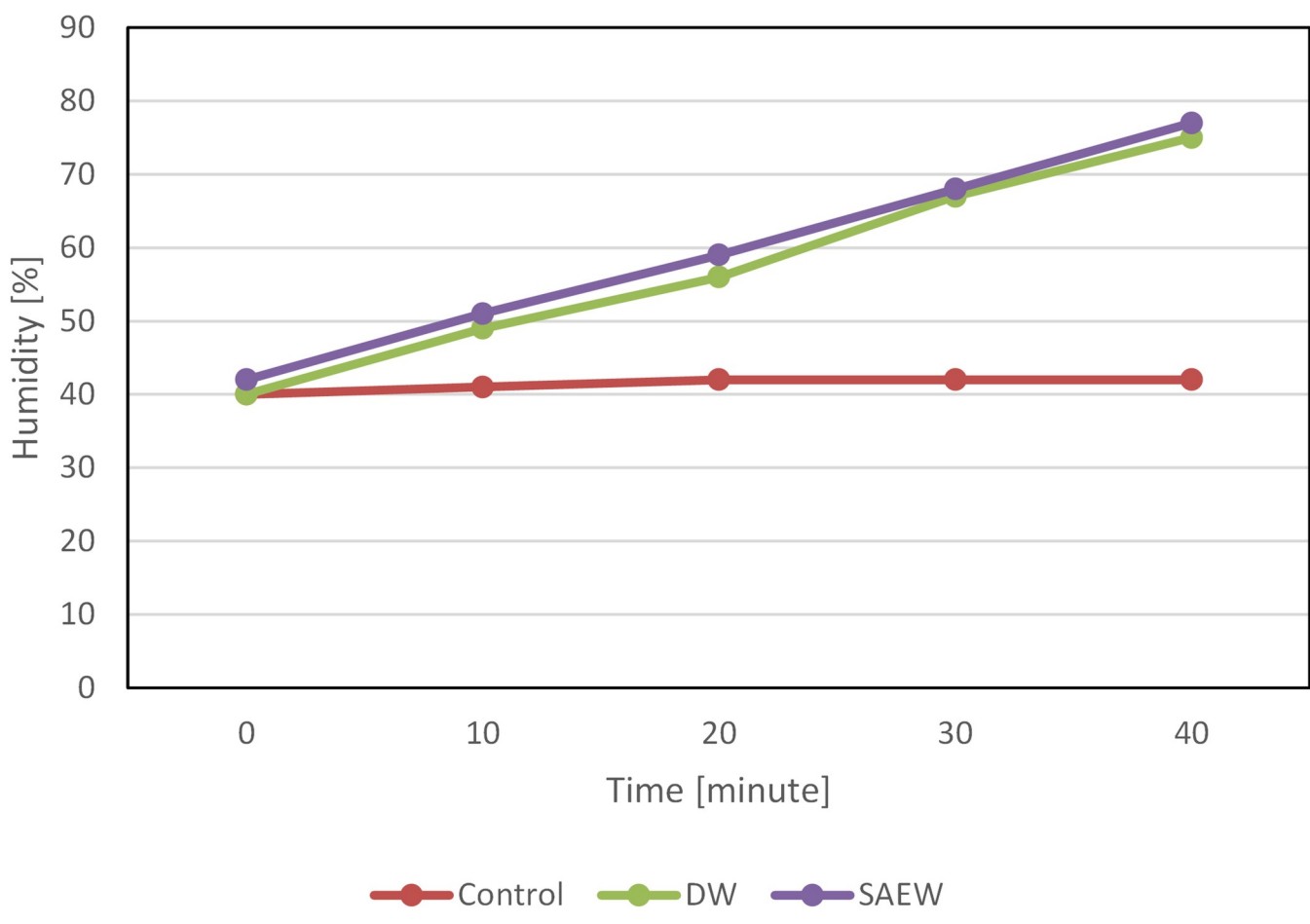

**Fig 3. Humidity inside the chamber.**

An aqueous solution (neutral electrolyzed water, or NEW) obtained by mixing the afore-mentioned acidic electrolyzed water in the anode chamber and alkaline water in the cathode chamber also has potent sterilizing action [15–17]. However, this is a mixture of 2 solutions, so it is unlikely to stabilize as the chemical reaction proceeds. SAEW was researched and developed by Japanese companies in the 1990s. In 1996, Morinaga Milk Industry Co. and Morinaga Engineering Co. applied for a patent on a method of production and production equipment, and the Japanese patent (JP 3798486) was registered in 2006. In contrast to AEW, SAEW has the same potent sterilizing action and is more stable [18].

Killing spores is extremely difficult because they are highly resistant to heat, so they often cause food poisoning. In the current study, SAEW with an available chlorine concentration of about 30 mg/L was able to kill *Bacillus* spores. If spores attached to food ingredients can readily be eliminated by SAEW beforehand, then this should reduce food poisoning.

Measures are needed to deal with airborne microorganisms in spaces occupied by people, such as food factories, waiting areas in hospitals, in airplanes, and greenhouses. In addition, reducing airborne viruses in pens and coops at hog and poultry farms is also a major need. Spatial sterilization is generally performed using ozone. However, ozone is highly toxic, and the allowable concentration in working environments is 0.05–0.1 ppm in most countries. Thus, sterilization with air containing ozone involves isolating humans and disinfected spaces and passing air through removal equipment after ozone sterilization, making the process

expensive. An air purifier is a system that sucks in air, purifies it via filtration, disinfection, or some other method, and then released purified air. This all takes time. Thus, the strong sterilization performance without need for air purification afterwards is sufficient reason to explore the potential for use of SAEW in aerial disinfection.

There are no systematic or standardized methods for assessing the microbiological effects of agents dispersed in air. One index can be found in the Methods for Assessing and Testing the Removal of Airborne Viruses by an Air Purifier devised by the Japan Electrical Manufacturers' Association (2011). According to that document, an effective removal is when the amount of the logarithmic decrease is greater than or equal to 2.0. In the current experiments, that standard served as a reference while being mindful of that airborne microorganisms were being eradicated. A logarithmic decrease of 3.43 was noted in just 20 min. Even after subtracting the amount attenuated by sterile water, SAEW was sufficiently effective in killing airborne microorganisms. There are few studies on the disinfection mechanism of SAEW. Ding et al. [19] found that SAEW disrupted permeability of the cell membrane and the cytoplasmic ultrastructure in *S. aureus* cells. Kim at al. [20] studied cell morphological states and cell permeability through a transmission electron microscope and found that SAEW could penetrate bacterial cell walls and induce cell damage and disruption. Wigginton el al. [21] identified the virus inactivation mechanisms as free chlorine (FC), reporting that FC caused losses in both genome replication and protein-mediated functions. They suggested that FC altered the capsid structure which facilitates access to protein structures that are inside.

SAEW has been sprayed at numerous sites such as hospitals, care facilities, kindergartens, and greenhouses in Japan over the past few years [8, 22], with no reports of health incidents due to the spraying. There are no statistically standardized methods of generally assessing the safety of spraying an antiseptic solution. COVID-19 is not yet under control, and SAEW is likely to be a way to help deal with that pandemic. Research on SAEW, including its safety, needs to be accelerated in the future.

## 5 Conclusions

Issues with the safety of sodium hypochlorite have been noted over the past few years. SAEW has gained attention as a replacement disinfectant, so the current experiment examined its sterilizing action. Results indicated that SAEW with an available chlorine concentration of about 30 mg/L has sufficient sterilizing action, even able to kill *Bacillus* spores within 15 minutes. *E. coli*, *P aeruginosa*, *S. enterica*, *S. aureus*, and *S. epidermidis* were eliminated within 15 sec of 30 mg/L of SAEW exposure. In addition, *C. cladosporioides* was removed within 60 sec of exposure to SAEW of 30 mg/L. Spatial spraying of SAEW with an available chlorine concentration of about 30 mg/L significantly disinfected a space the size of a small room in a short period of time.

## Supporting information

**S1 File.**
(XLSX)

## Acknowledgments

The authors would also like to thank the anonymous reviewer for their valuable comments.

## Author Contributions

**Conceptualization:** Midori Kurahashi, Takaaki Ito.

**Data curation:** Midori Kurahashi, Angelica Naka.

**Formal analysis:** Midori Kurahashi.

**Investigation:** Midori Kurahashi, Angelica Naka.

**Methodology:** Midori Kurahashi.

**Project administration:** Midori Kurahashi.

**Resources:** Midori Kurahashi, Takaaki Ito.

**Supervision:** Midori Kurahashi.

**Validation:** Midori Kurahashi, Takaaki Ito, Angelica Naka.

**Visualization:** Midori Kurahashi, Angelica Naka.

**Writing – original draft:** Midori Kurahashi, Angelica Naka.

**Writing – review & editing:** Angelica Naka.

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
