## [Decision Letter · Decision Letter 0]

4 Feb 2021

PONE-D-20-33738

Spatial disinfection potential of slightly acidic electrolyzed water

PLOS ONE

Dear Dr. Naka,

Thank you for submitting your manuscript to PLOS ONE. After careful consideration, we feel that it has merit but does not fully meet PLOS ONE’s publication criteria as it currently stands. Therefore, we invite you to submit a revised version of the manuscript that addresses the points raised during the review process.

In addition to review comments, you failed to disclose any potential conflict of interest with the SAEW solution used your study. Please ensure to include the product number of the SAEW used, if it is commercially available, otherwise, provide details pertaining to SAEW formulation.  Funding details are also not disclosed, please add this information in the acknowledgment section. 

When reviewing the experimental design dealing with indoor spraying of SAEW, no information was provided as to why *S. epiderrmidis *was the only selected organism for this study. Why were other previously investigated species not tested in this assay, considering that outcomes may vary from the use of SAEW directly on pure culture or indirectly via indoor spraying? 

Finally, the antimicrobial potency of SAEW in your study was established based on CFU counts. Have you looked into other methods for validating cell death? 

We look forward to receiving your revised manuscript.

Kind regards,

Olivier Habimana

Academic Editor

PLOS ONE

Journal Requirements:

"The authors have declared that no competing interests exist"

We note that one or more of the authors are employed by a commercial company: Organo Food Tech Corporation.

(2) Please also provide an updated Competing Interests Statement declaring this commercial affiliation along with any other relevant declarations relating to employment, consultancy, patents, products in development, or marketed products, etc.  

Reviewers' comments:

Reviewer's Responses to Questions

**Comments to the Author**

1. Is the manuscript technically sound, and do the data support the conclusions?

Reviewer #1: Partly

2. Has the statistical analysis been performed appropriately and rigorously? 

Reviewer #1: I Don't Know

3. Have the authors made all data underlying the findings in their manuscript fully available?

Reviewer #1: Yes

4. Is the manuscript presented in an intelligible fashion and written in standard English?

Reviewer #1: Yes

5. Review Comments to the Author

Reviewer #1: The authors submitted a document about the use of slightly acidic electrolyzed water on spatial disinfection. It is a very relevant document however, there are some points that need to be clarified.

L42. Authors stablished that sodium hypochlorite is highly safe however in some European countries, its use is banned. It will need to be modify.

L74. The manuscript stablished: “Spatial disinfection with a conventional agent involves spraying in an unoccupied space or using protective equipment. Spatial disinfection can be performed with a low concentration of SAEW” At the end of the paragraph, authors cited this document https://doi.org/10.1637/11107-042115-Reg.1where it is stablished that Newcastle virus is affected by the use of higher concentration of SAEW (100 ppm). Authors need to clarify its statement.

L80. It is a little confuse because authors wrote about the impact of multidrug-resistant microorganisms and the necessity for big area disinfection like food factories, hospitals,

care facilities, kindergartens, and animal pens and coops. So, it is implying that the study will focus about airborne pathogens. But, at L94, most of the pathogens are not airborne. I believe authors needs to justify it statement why they are using bacteria like Salmonella enterica, Staphylococcus aureus and/or E. coli.

L172. It is confused when in text authors use CFU/mL, then on L174 log10 CFU/mL. Please, use only one type of units. I recommend using log10 CFU/mL.

L276. There is no reference on the use of EW on those facilities.

Authors worked with S. epidermis, which causes biofilms. Results are interesting however, I suggest performing an in vitro study with this bacteria and to sample walls of the chamber where indoor study was performed. This will provide extra information about the survival/escaped bacteria after EW air treatment.

Because SAEW is not a neutral water, its pH could affect/damage different surfaces. It could be interesting to perform a corrosion test.

6. PLOS authors have the option to publish the peer review history of their article (what does this mean?). If published, this will include your full peer review and any attached files.

Reviewer #1: No

---

## [Author Response · Author response to Decision Letter 0]

25 Mar 2021

We would like to thank the editor and the reviewers for their time to read our manuscript PONE-D-20-33738 “Spatial disinfection potential of slightly acidic electrolyzed water” and giving us valuable comments and suggestions to improve its quality. We have revised the manuscript accordingly. Our responses to the editor and reviewers’ comments are as follows.

-----------------

<1.1> In addition to review comments, you failed to disclose any potential conflict of interest with the SAEW solution used your study. Please ensure to include the product number of the SAEW used, if it is commercially available, otherwise, provide details pertaining to SAEW formulation. Funding details are also not disclosed, please add this information in the acknowledgment section. 

<1.1 Response>

We would like to thank the editor for the valuable comments and suggestions. We conducted this research with the annual budget of the University of Tokyo. However, this budget was to conduct several research projects, not specifically to work on SAEW research. Following the editor's advice (by email), we left the Acknowledgements section unchanged:

Lines 309-310

“The authors would also like to thank the anonymous reviewer for their valuable comments.”

We discussed experimental design and results (conceptualization and validation) with Organo Food Tech Corporation. They also provided the SAEW and the clean chamber for the spatial disinfection test (resources). Regarding SAEW, we added the following information:

Lines 128-129

“Slightly acidic electrolyzed water (HOCL0.36 t, HOCL Inc.) was specifically prepared by Organo Food Tech, Saitama, Japan for the purpose of these experiments.”

-----------------

<1.2> When reviewing the experimental design dealing with indoor spraying of SAEW, no information was provided as to why S. epidermidis was the only selected organism for this study. Why were other previously investigated species not tested in this assay, considering that outcomes may vary from the use of SAEW directly on pure culture or indirectly via indoor spraying? 

<1.2 Response>

We conducted spatial disinfection experiments only with S. epidermidis because it is generally regarded as a safe microorganism that is present in human skin and usually does not cause pyogenic infections unless there are compromised patients. One of the objectives of this research is to report on the efficacy of SAEW when used as spray (mist). Considering that we do not have a proper infrastructure to spray pathogenic and infectious microorganisms, we selected S. epidermidis for spatial disinfection experiments. We added the following sentence:

Lines 122-125

“S. epidermidis was the only microorganism selected for spatial disinfection experiments because it is a common bacterium present in the human skin and, thus, does not usually cause infections in non-compromised patients and does not require high level of biosafety laboratories.”

To show the effectiveness of SAEW on other pathogenic bacteria, we conducted experiments on pure culture.

-----------------

<1.3> Finally, the antimicrobial potency of SAEW in your study was established based on CFU counts. Have you looked into other methods for validating cell death?

<1.3 Response>

No, we have not investigated other methods for validating cell death. We used CFU count method because we found that this is one of the most common methods used by research teams that worked on research related to bacterium removal efficacy.

-----------------

<2.1> The authors submitted a document about the use of slightly acidic electrolyzed water on spatial disinfection. It is a very relevant document however, there are some points that need to be clarified. L42. Authors stablished that sodium hypochlorite is highly safe however in some European countries, its use is banned. It will need to be modify.

<2.1 Response>

The authors would like to thank the reviewer for the feedback and suggestions to improve our manuscript. 

The reviewer is right, even though the use of sodium hypochlorite is still allowed in some countries such as Japan and Korea, its use is banned in many countries. So, we modified the sentence as follows:

Lines 42-44

“Sodium hypochlorite solutions are inexpensive and relatively effective, so they have been used around the world for over 100 years.”

-----------------

<2.2> L74. The manuscript stablished: “Spatial disinfection with a conventional agent involves spraying in an unoccupied space or using protective equipment. Spatial disinfection can be performed with a low concentration of SAEW” At the end of the paragraph, authors cited this document https://doi.org/10.1637/11107-042115-Reg.1where it is stablished that Newcastle virus is affected by the use of higher concentration of SAEW (100 ppm). Authors need to clarify its statement.

<2.2 Response>

The reviewer is right, Newcastle virus was affected by 50 and 100 mg/L of free chlorine concentration. SAEW disinfection effectiveness depends on the type and concentration of the microorganisms. It also depends on the SAEW’s free chlorine concentration and the sprayed volume and duration. Unfortunately, there are very few experiments on spatial disinfection effectivity of SAEW. We added references of research works that were conducted with 30 mg/L and 80 mg/L of free chlorine concentration. Please refer to the following paragraph:

Lines 77-82

“Spatial disinfection can be performed spraying SAEW with free chlorine concentrations of 100 mg/L or lower and without concern for residual chlorine. There are numerous studies on the sterilizing action of electrolyzed water, but there are very few studies on the use of SAEW for indoor disinfection [4-6].”

-----------------

<2.3> L80. It is a little confuse because authors wrote about the impact of multidrug-resistant microorganisms and the necessity for big area disinfection like food factories, hospitals, care facilities, kindergartens, and animal pens and coops. So, it is implying that the study will focus about airborne pathogens. But, at L94, most of the pathogens are not airborne. I believe authors needs to justify it statement why they are using bacteria like Salmonella enterica, Staphylococcus aureus and/or E. coli.

<2.3 Response>

The reviewer is correct, not all species tested in this research are airborne bacteria. 

One of the objectives of this research is to evaluate the efficacy of SAEW against bacteria when applied as spray (mist). Unfortunately, our laboratory does not have a proper infrastructure to spray pathogenic and infectious bacteria. So, we conducted spray experiments only with S. epidermidis which is generally regarded as a safe microorganism unless there are compromised patients. To evaluate the effectiveness of SAEW against pathogenic bacteria and bacterial spores, we applied SAEW directly to pure culture (we can handle them in our biological safety cabinet).

We added the following paragraph:

Lines 77-82

“Its efficacy as disinfectant was measured in this research by directly applying SAEW on different types of bacteria and bacterial spores. Spatial disinfection can be performed spraying SAEW with free chlorine concentrations of 100 mg/L or lower and without concern for residual chlorine. There are numerous studies on the sterilizing action of electrolyzed water, but there are very few studies on the use of SAEW for indoor disinfection [4-6].”

-----------------

<2.4> L172. It is confused when in text authors use CFU/mL, then on L174 log10 CFU/mL. Please, use only one type of units. I recommend using log10 CFU/mL.

<2.4 Response>

The reviewer is right. We corrected the mistake as follows:

Lines 176-177

“Treatment of B. cereus (spores), a gram-positive bacillus that causes food to spoil, resulted in 1/10 of the original log10 CFU/mL after 60 sec.”

-----------------

<2.5> L276. There is no reference on the use of EW on those facilities.

<2.5 Response>

Following the reviewer’s suggestion, we added the references accordingly.

Lines 288-290

“SAEW has been sprayed at numerous sites such as hospitals, care facilities, kindergartens, and greenhouses in Japan over the past few years [8, 22], with no reports of health incidents due to the spraying.”

-----------------

<2.6> Authors worked with S. epidermidis, which causes biofilms. Results are interesting however, I suggest performing an in vitro study with this bacteria and to sample walls of the chamber where indoor study was performed. This will provide extra information about the survival/escaped bacteria after EW air treatment. Because SAEW is not a neutral water, its pH could affect/damage different surfaces. It could be interesting to perform a corrosion test.

<2.6 Response>

We performed two type of experiments:

- In vitro experiments with pathogenic and infectious bacteria and bacterial spores.

- Spray experiment with S. epidermidis, regarded as a safe microorganism (it is present in the human skin) and usually does not cause pyogenic infections unless there are compromised patients.

The reviewer is correct, some bacteria may adhere to the walls. However, considering that SAEW’s free chlorine concentration is 41 ppm (this means that 99.59% is water), we believe that SAEW spray is not suitable for surface disinfection. For surface disinfection, it is more effective if we apply SAEW directly and in great amount. In this experiment we focused on bacteria that were suspended in the chamber and tested both SAEW and water, the latter as control. Hypochlorous acid (HOCl), which is the active molecule in SAEW, easily reacts with organic matter. Thus, if applied as spray (considering that HOCl concentration in SAEW is low), it will probably react with some dust on the surface and not with the bacteria.

Regarding experiments with slightly acidic water, it could be interesting, but we believe that pH 6 may not have great impact on bacteria tested in this research. These bacteria usually show protection mechanisms against acid stress and overcome the challenge posed by different acidic environments.

-----------------

<3.1> L101 – 102 …. on nutrient agar (Eiken Chemical Co.) and B. subtilis was cultured on soybean-casein digest agar (Eiken Chemical…

L111. … was cultured on potato dextrose agar….

L118 ….. cultured in a heart infusion broth…

L133… soybean-casein digest medium

L152 – 153 ….ultrasonic humidifier..

Please add the code number of all used media and machine/equipment not only the brand or which company they are form.

<3.1 Response>

The reviewer is correct. We followed the reviewer suggestion and added the necessary information.

Lines 101-102

“B. cereus was cultured on nutrient agar PT2810 (Eiken Chemical Co.) and B. subtilis was cultured on soybean-casein digest agar PT8010 (Eiken Chemical Co.)”

Lines 108-109

“E. coli, P. aeruginosa, S. enterica, and S. aureus were each cultured on nutrient agar PT 1010 (Eiken Chemical Co.) at 35℃±1℃ for 18–24 h.”

Lines 111-113

“C. cladosporioides was cultured on potato dextrose agar PT 2610 (Eiken Chemical Co.) at 25℃±1℃ for 7–10 days.”

Lines 118-120

“The test bacterium was cultured in a brain heart infusion broth PT3025 (Eiken Chemical Co.) at 35℃±1℃ for 18–24 h.”

Lines 136-145

“This test solution was stored at 20℃±1℃. After 15 s, 1 min, and 15 min, it was immediately diluted 10-fold with soybean-casein digest medium with polysorbate 80 & lecithin SCDLP “DAIGO” (Nihon Pharmaceutical Co.). Microorganisms in the test solution were counted using a medium for cultivation and isolation. The medium for cultivation and isolation was soybean-casein digest agar with polysorbate 80 & lecithin SCDLP “DAIGO” (Nihon Pharmaceutical Co.) for B. cereus, B. subtilis, E. coli, P. aeruginosa, S. enterica, and S. aureus. Each bacterium was then cultured at 35℃±1℃ for 2 days using the pour plate method. The medium for cultivation and isolation was glucose peptone agar with lecithin & polysorbate 80 GPLP AGAR “DAIGO” (Nihon Pharmaceutical Co.) for C. cladosporioides.”

Lines 155-158

“Three mL of this mixture (2 mL of the bacterial solution + 1 mL of PBS) was sprayed at a rate of 1 mL/min for 3 min from the spray nozzle of an ultrasonic humidifier AHD-012 (Shizuku Plus+, Apix International, Osaka, Japan) in a large clean chamber.” 

-----------------

<3.2> L 126 – 128. SAEW used in the experiment involving pure cultures had an available chlorine concentration of 30 mg/kg and a pH of 5.9. SAEW used in the spatial disinfection experiment had an available chlorine concentration of 41 mg/kg and a pH of 6.0. Why the available chlorine concentration used for pure culture and in spatial disinfection experiment were different?

<3.2 Response>

We conducted spatial disinfection experiments as soon as we received the SAEW from Organo Food Tech Corporation. At that time, the concentration of free chlorine was 41 mg/L. Some weeks later, we conducted experiments on pure culture. At that time, the concentration was 30 mg/L (free chlorine concentration decreased with time, but it was still effective). We measured free chlorine concentration just before experiments.

-----------------

<3.3> 2.3. Effects of SAEW on pure cultures. How many replication samples you’ve done for each microorganism?

<3.3 Response>

We performed experiments in triplicate. We added this information in the manuscript. Please refer to Line 148. 

Line 148

“All experiments were performed in triplicate.”

-----------------

<3.4> 2.4. Effects of indoor spraying of SAEW. How many replications process (not sampling time) you’ve done in this study?

<3.4 Response>

We performed indoor spraying experiments three times. We added this information in the manuscript. Please refer to Line 163. 

Line 163

“This experiment was repeated three times.”

-----------------

<3.5> Statistical analysis. Please provide information about the statistical tool that been used.

<3.5 Response>

The obtained results were statistically assessed using Tukey’s honestly significant difference (HSD) test. Please refer to Lines 167-169. 

Lines 167-169

“The obtained results were statistically assessed using Tukey’s honestly significant difference (HSD) test, with P < 0.05 considered to be statistically significant.”

-----------------

<3.6> Discussion. The discussions are not deep enough. I can’t see the discussion of the mechanism on how the SAEW can inhibit or reduce the microbial growth in the studies. 

<3.6 Response>

The reviewer is right. We added the following paragraph regarding SAEW’s bacteria inhibition mechanism.

Lines 278-286

“There are few studies on the disinfection mechanism of SAEW. Ding et al. [19] found that SAEW disrupted permeability of the cell membrane and the cytoplasmic ultrastructure in S. aureus cells. Kim at al. [20] studied cell morphological states and cell permeability through a transmission electron microscope and found that SAEW could penetrate bacterial cell walls and induce cell damage and disruption. Wigginton el al. [21] identified the virus inactivation mechanisms as free chlorine (FC), reporting that FC caused losses in both genome replication and protein-mediated functions. They suggested that FC altered the capsid structure which facilitates access to protein structures that are inside.”

-----------------

<3.7> General comments 

- The manuscript is well written. However, some details are missing. 

- It is crucial to give the detail of all medias and used equipment/tools/machines being used. This information is needed to verify your methods and enable the reader mimicking your study. 

- The discussions are not deep enough. I can’t see the discussion of the mechanism on how the SAEW can inhibit or reduce the microbial growth in the studies. 

<3.7 Response>

We would like to thank the reviewer for the feedback. We revised the manuscript according to the reviewer’s comments and suggestions.

-----------------

---

## [Decision Letter · Decision Letter 1]

20 Apr 2021

PONE-D-20-33738R1

Spatial disinfection potential of slightly acidic electrolyzed water

PLOS ONE

Dear Dr. Naka,

Thank you for submitting your manuscript to PLOS ONE. After careful consideration, we feel that it has merit but does not fully meet PLOS ONE’s publication criteria as it currently stands. Therefore, we invite you to submit a revised version of the manuscript that addresses the points raised during the review process.

We look forward to receiving your revised manuscript.

Kind regards,

Olivier Habimana

Academic Editor

PLOS ONE

Journal Requirements:

Reviewers' comments:

Reviewer's Responses to Questions

**Comments to the Author**

1. If the authors have adequately addressed your comments raised in a previous round of review and you feel that this manuscript is now acceptable for publication, you may indicate that here to bypass the “Comments to the Author” section, enter your conflict of interest statement in the “Confidential to Editor” section, and submit your "Accept" recommendation.

Reviewer #1: (No Response)

2. Is the manuscript technically sound, and do the data support the conclusions?

Reviewer #1: Yes

3. Has the statistical analysis been performed appropriately and rigorously? 

Reviewer #1: Yes

4. Have the authors made all data underlying the findings in their manuscript fully available?

Reviewer #1: Yes

5. Is the manuscript presented in an intelligible fashion and written in standard English?

Reviewer #1: Yes

6. Review Comments to the Author

Reviewer #1: I previously suggested to authors to perform an in vitro study with S. epidermidis, authors answer me back the following statement: we believe that pH 6 may not have great impact on bacteria tested in this research. These bacteria usually show protection mechanisms against acid stress and overcome the challenge posed by different acidic environments.

It is not clear why authors make that statement if results show that spatial disinfection worked properly. I suggest including an in vitro study with S. epidermidis.

7. PLOS authors have the option to publish the peer review history of their article (what does this mean?). If published, this will include your full peer review and any attached files.

Reviewer #1: **Yes: **Jose Alberto Cano-Buendia

---

## [Author Response · Author response to Decision Letter 1]

3 Jun 2021

<1.1>

I previously suggested to authors to perform an in vitro study with S. epidermidis, authors answer me back the following statement: we believe that pH 6 may not have great impact on bacteria tested in this research. These bacteria usually show protection mechanisms against acid stress and overcome the challenge posed by different acidic environments. It is not clear why authors make that statement if results show that spatial disinfection worked properly. I suggest including an in vitro study with S. epidermidis.

<Response 1.1>

We would like thank the reviewer for the valuable comment and suggestion. As suggested by the reviewer, we performed in vitro experiments with S. epidermidis and added the results in the manuscript. Please refer to Lines 94-100, 113-116, 144-146, 187-189, 193-196 and 309-310 and Figure 1.

Lines 94-100

An in vitro experiment was performed with Bacillus cereus NBRC 13494, Bacillus subtilis NBRC 3134, Pseudomonas aeruginosa NBRC 13275, Salmonella enterica subsp. enterica NBRC 3313, Staphylococcus aureus subsp. aureus NBRC 12732, Cladosporium cladosporioides NBRC 6348, and Staphylococcus epidermidis NBRC 12993 from the NITE Biological Resource Center (NBRC, Japan) and Escherichia coli ATCC 43895 (serotype O157:H7, a verotoxin I and II-producing strain) from the American Type Culture Collection (ATCC, USA). 

Lines 113-116

S. epidermidis was cultured in a brain heart infusion broth PT3025 (Eiken Chemical Co.) at 35℃±1℃ for 18–24 h. The bacterial count was adjusted to 107-108 CFU/mL with a sterile phosphate buffer (phosphate-buffered saline, or PBS) to yield the test bacterial solution.

Lines 144-146

The medium for cultivation and isolation was soybean-casein digest agar with polysorbate 80 & lecithin SCDLP “DAIGO” (Nihon Pharmaceutical Co.) for B. cereus, B. subtilis, E. coli, P. aeruginosa, S. enterica, S. aureus, and S. epidermidis

Lines 187-189

Treatment of S. aureus and S. epidermidis, gram-positive facultative anaerobic bacilli, with electrolyzed water resulted in logarithmic (log10 CFU/mL) decrease of 5.2 or greater and 5.4 or greater, respectively within 15 sec.

Lines 193-196

Fig 1. Effects of SAEW on pure cultures (a) Bacillus cereus, (b) Bacillus subtilis, (c) Escherichia coli (O157:H7), (d) Pseudomonas aeruginosa, (e) Salmonella enterica subsp. enterica, (f) Staphylococcus aureus subsp. aureus, (g) Cladosporium cladosporioides, and (h) Staphylococcus epidermidis.

Line 307-310

Results indicated that SAEW with an available chlorine concentration of about 30 mg/L has sufficient sterilizing action, even able to kill Bacillus spores within 15 minutes. E. coli, P aeruginosa, S. enterica, S. aureus, and S. epidermidis were eliminated within 15 sec of 30 mg/L of SAEW exposure.

As for the statement regarding pH, the reviewer is correct, this does not answer the reviewer’s question. We wanted to mention that the disinfectant capacity of SAEW is attributed to the presence of HOCl.

---

## [Editor Report · Decision Letter 2]

9 Jun 2021

Spatial disinfection potential of slightly acidic electrolyzed water

PONE-D-20-33738R2

Dear Dr. Naka,

We’re pleased to inform you that your manuscript has been judged scientifically suitable for publication and will be formally accepted for publication once it meets all outstanding technical requirements.

Kind regards,

Olivier Habimana

Academic Editor

PLOS ONE
---

## [Editor Report · Acceptance letter]

24 Jun 2021

PONE-D-20-33738R2 

Spatial disinfection potential of slightly acidic electrolyzed water 

Dear Dr. Naka:

I'm pleased to inform you that your manuscript has been deemed suitable for publication in PLOS ONE. Congratulations! Your manuscript is now with our production department. 

Kind regards, 

on behalf of

Dr. Olivier Habimana 

Academic Editor

PLOS ONE